# Reducing Effect of Farnesylquinone on Lipid Mass in *C. elegans* by Modulating Lipid Metabolism

**DOI:** 10.3390/md17060336

**Published:** 2019-06-05

**Authors:** Xihua Jia, Manglin Xu, Aigang Yang, Yan Zhao, Dong Liu, Jian Huang, Peter Proksch, Wenhan Lin

**Affiliations:** 1State Key Laboratory of Natural and Biomimetic Drugs, Peking University, Beijing 100191, China; jiaxihua_glia@163.com (X.J.); hyldxmldyx@163.com (M.X.); yangaifang88@163.com (A.Y.); young_106@163.com (Y.Z.); liudong_1982@126.com (D.L.); jhuang@bjmu.edu.cn (J.H.); 2Institute für Pharmazeutische Biologie und Biotechnologie, Heinrich-Heine-Universität Düsseldorf, 40225 Düsseldorf, Germany; proksch@uni-duesseldorf.de

**Keywords:** farnesylquinone, nitrosporeunol H, *C. elegans*, β-oxidation, lipid-lowering effect, fatty acid profile

## Abstract

Bioassay-guided fractionation of marine-derived fungi revealed that the EtOAc fraction from the fermentation broth of a mutated fungal strain *Streptomyces nitrosporeus* YBH10-5 had lipid-lowering effects in HepG2 cells. Chromatographic separation of the EtOAc fraction resulted in the isolation of 11 PKS-based derivatives, including a structurally unique meroterpenoid namely nitrosporeunol H (**1**). The structure of compound **1** was determined by the analysis of spectroscopic data. Further bioassay resulted in farnesylquinone (**2**) and its analogues to exert in vivo fat-reducing effects in *C.*
*elegans* worm model. The underlying mode of action of compound **2** in the context of live worms was investigated, uncovering that compound **2** enhanced the mitochondrial β-oxidation rate and changed the transcriptional level of energy metabolism genes. Additional experiments revealed that compound **2** exerted its effects in *C. elegans* partially through repressing FAT-5, an isoform of stearoyl-CoA desaturase (SCD) which catalyzes the conversion of saturated fatty acids to monounsaturated fatty acids, thereafter leading to the modification of the fatty acid profile. Thus, compound **2** was suggested to be a promising lead for further optimization to treat obesity.

## 1. Introduction

Metabolic diseases, such as diabetes, obesity, and atherosclerosis caused serious problems in global human health [1]. Metabolic disorders are induced by the dysfunction of metabolism and/or immunity. Obesity is a related metabolic disorder that has been a serious threat to human health worldwide with an increased incidence of metabolic diseases [2]. It is caused by an excess of adipose tissue and an imbalance between energy intake and expenditure and is also attributed to the diet, genetic disorders, sedentary lifestyle, and psychological factors, that increases the risk of developing type 2 diabetes, hypertension, and cardiovascular diseases [3]. Lipid metabolic disorders are also the pathological basis of atherosclerosis, which is a deleterious condition caused by the deposition of foam cells and extracellular materials mainly in arteries. Although pharmaceutical industries are striving to provide more options for patients with diabetes, hypertension, and dyslipidemia, of which obesity is considered to be the main driver. Anti-obesity drugs are few in number due to the complex etiology of obesity [4,5,6,7]. Thus, finding new leads that can alleviate this metabolic disorder is currently urgent. Natural products provided as a promising source for the discovery of lead compounds to combat obesity and its related diseases. Flavonoids and phenols have been reported to be effective in the treatment of obesity [8,9,10,11,12]. For the bioassay to discover bioactive natural compounds with lipid-lowering effects, nematode *Caenorhabditis elegans* has been successfully used as an animal model to study energy metabolism, identify regulatory genes relevant to lipid dynamic changes in forward genetic screens, and to seek compounds with lipid modulation activity [13,14,15]. The core molecular mechanisms to regulate lipid metabolism are well conserved across phyla. More than 400 genes in *C. elegans* directly attend lipid synthesis and catabolism, and play the roles as pivotal energy sensors or foraging regulators, while the energy mobilization and fat storage have counterparts in mammals [13]. In our previous work, a number of marine-derived compounds such as farnesylquinone (**2**) and related analogs showed remarkably decreasing lipid levels in HepG2 cells [16]. In present work, the mode of action of compound **2** in intact *C. elegans* was investigated to uncover whether compound **2** showed the relevant effect in the animal model and influenced energy homeostasis, the expression of metabolic genes and fatty acid composition.

## 2. Results

### 2.1. Structure Identification

In order to supply sufficient samples for further investigation of the in vivo activity and the mode of action for the lipid-lowering activities, a large scale fermentation of the mutated fungal strain of Arctic *Streptomyces nitrosporeus* YBH10-5 was performed for the re-isolation of the active compounds. Chromatographic separation of the EtOAc extract from the cultured fungus resulted in the isolation of 11 compounds, including a new compound namely nitrosporeunol H (**1**) (Figure 1).

Nitrosporeunol H (**1**) was isolated as a purplish red amorphous. Its molecular formula was determined to be C_44_H_56_O_6_ based on the high resolution electrospray ionization mass spectroscopy (HRESIMS) (*m/z* 647.4106 [M + H]^+^) (Appendix A) and NMR data, requiring 17 degrees of unsaturation. The IR absorptions at 3446, 1653 and 1629 cm^−1^ suggested the presence of hydroxy and unsaturated ketone groups (Appendix A). The ^1^H NMR spectrum (Appendix A) exhibited ten methyl, six olefinic, two phenyl protons, and a phenol proton, as well as a number of methylene protons. The 1D and 2D NMR data (Table 1, Appendix A) established a partial structure for a dihydro-1,4-anthracenedione nucleus, as evident from the olefinic proton H-3 (δ_H_ 6.52, s) showing the HMBC correlations with the unsaturated ketones C-4 (δ_C_ 187.5) and C-1 (δ_C_ 186.4) and C-5 (δ_C_ 130.7) and from the methyl protons H_3_-7 (δ_H_ 2.04, s) to C-2 (δ_C_ 144.9), C-3 (δ_C_ 134.3), and C-4 for a 3-methylquinone unit; an aromatic proton H-3’ (δ_H_ 6.57, s) showing the heteronuclear multiple bond correlation (HMBC) with C-1’ (δ_C_ 144.4), C-4’ (δ_C_ 145.8) and C-5’ (δ_C_ 117.9), in addition to the HMBC correlations from H_3_-7’ (δ_H_ 2.13, s) to C-1’, C-2’ (δ_C_ 124.7) and C-3’ (δ_C_ 118.7) for a methylated penta-substituted phenyl ring. Additional HMBC correlations from H-8’ (δ_H_ 5.12, d, *J* = 12.0 Hz) to C-1, C-5’, C-6’ (δ_C_ 115.5), and C-4’ and from H-9 to C-5 and C-6’ connected the subunits between the aromatic ring and quinone unit and allowed the establishment of a nucleus structure. Further analysis of the ^1^H-^1^H correlation spectroscopy (COSY) and HMBC data afforded two linear side chains of acyclic sesquiterpenes, structurally related to trimethyldodecatriene. The distinction of both side chains was attributed to the structural modification at the terminal residue (Figure 2). In side chain A, the HMBC correlation of H_3_-19 (δ_H_ 1.31, s) to C-9 (δ_C_ 136.1), C-10 (δ_C_ 78.0) and C-11 (δ_C_ 41.8) clarified C-10 to be oxygenated, while C-8 of the terminal double bond linked to C-5 and C-6’. The COSY relationship between the olefinic proton H-9’ (δ_H_ 4.69, d, *J* = 12.0 Hz) and the methine proton H-8’ in association with the HMBC correlation of H-8’ with the carbon resonances in the aromatic ring and the quinone ring indicated that C-8’ connected C-5’ and C-6 (Figure 2). Calculation of the molecular unsaturation allowed the remaining site of unsaturation to be contributed by an ether bond, which was located between C-1’ and C-10. The double bonds in both side chains were assigned to *E* geometry based on the chemical shifts of the olefinic methyl carbons (<20 ppm) and the NOE interaction. Based on the electronic circular dichroism (ECD) rule for chromenes (Crabbe’s rule) [17,18], the Cotton effect at around 270 nm for the 1Lb excitation reflected the absolute configuration of the chiral center in chromene chromophore. The negative Cotton effect at 270 nm in compound **1** was in accordance with the chromene *P*-helicity (Figure 3), which was in agreement with *R* configuration. However, the absolute configuration of C-8’ was uncertain.

Biogenetically, farnesylquinone (2) was speculated to be the precursor to generate compound **1** via a dimerization. Firstly, an enol isomerization occurred by acidic induction to derive an intermediate **a**, while subsequent oxidative formal [3 + 3] cycloaddition of two molecules of intermediate **a** resulted in the dimerization. Tautomerization of intermediate **b** yielded a phenol **c**, which followed a 6π-cyclization to generate compound **1** (Scheme 1).

In addition, ten compounds were identical to farnesylquinone (**2**), farnesylbenezenediol (**3**), nitrosporeunol E (**4**), nitrosporeunol F (**5**), 7-deacetoxyyanuthone (**6**), nitrosporeunol G (**7**), 3,7-dimethyl- 2,6-octadienylbenezenediol (**8**), isonitrosporeunol F (**9**), geranyllinalool (**10**), and 2-methylquinone (**11**) (Figure 1), based on the comparison of their spectroscopic data and specific rotation with those of the authentic samples [16].

### 2.2. Compounds Induced Nile Red Phenotypes of Worms

*C. elegans* (worm) deposits its fat mainly in the intestine, but it lacks adipocytes that are found in all mammals. Nile red stained fat droplets are easily visualized with fluorescence microscopy due to *C. elegans*’ transparent bodies. Thus, worms *C. elegans* have been widely used to identify and to characterize metabolic pathways involved in fat metabolism [19,20]. In preliminary experiments, all compounds were tested for the effects to regulate fat in *C. elegans* based on the Nile red fluorescence intensity that represents lipid droplets in N2 worms. It is noteworthy that the quinone nucleus or its phenol isomer with a linear sesquiterpene chain such as compounds **2** and **3** significantly reduced the fat density (Figure 4). However, the quinone core as the case of 2-methylquinone (**11**) or the linear diterpene (**10**) weakly impacted on the fat mass in worms. These findings suggested that the linear sesquiterpene but not alone is necessary for the quinone derivatives to increase the inhibitory effects. The quinone nucleus bearing a linear monoterpene instead of the sesquiterpene side chain is of key determinant of the fat-reducing effect, as exemplified by compound **8** which was dramatically less potent in fat-lowing phenotypes in comparison with that of compound **3**. The chromame or chromene meroterpenoids (compounds **4**, **5** and **9**) showed weaker effects in the worm model than those of compounds **2** and **3**. These data in association with the relative fluorescent density of **7**-treated worms being 0.92 compared to 0.59 in the **3**-treated worms suggested that the structure modification at side chain of compound **2** or **3** was unable to increase the lipid-lowering activity. Oxidation of the quinone core such as compound **6** led to the disappearance of the Nile red phenotype, implying that quinone or its phenol ring is the essential core to play the lipid-lowering role. Compound **1**, a dimer of compound **2**, showed weaker activity for fat modulating in comparison with that of compound **2**.

The relationship between the dose and activity of compound **2** was further analyzed by a solvatochromatic vital dye with the Nile red in the *C. elegans* model. The reduction of Nile red fluorescence intensity in worms showed a dose-dependent manner after treatment with compound **2** (Figure 5A,B), while compound **2** in a dose of 0.5 mM decreased approximately 50% Nile red fluorescence intensity compared to control. The maximum lipid decreasing effect was observed when the dose of compound **2** increased to 1.0 mM, it made the worms having only 31.9% Nile red stained lipid left (Figure 5B). To further ascertain the fat phenotype as induced by compound 2 to be related to its lipid-lowering effects, the triglyceride level of **2**-treated worms was measured. The total amount of triglyceride was decreased by 24% compared to that of the DMSO control (Figure 5C). This data was consistent with the Nile red staining results, supporting that compound **2** is able to reduce lipid in *C. elegans*. Alternatively, the Oil red-O stained HepG2 cells, the mammalian model cells which exposed to albumin-conjugated oleic acid leading lipid droplets accumulated inside the cells, were treated by compound **2**. The results showed that compound **2** obviously alleviated steatosis in HepG2 cells (Figure 5D), indicating that the action of compound **2** on *C. elegans* is relevant to the mammalian system.

### 2.3. Compound ***2*** Had No Detrimental Effect on the Health of Worms

To investigate whether the variation of Nile red phenotype induced by compound **2** was attributed to the general detrimental effects on *C. elegans*, the developmental phenotypes and behavioral phenotype were examined. Worms exposed to 1.0 mM of compound **2** showed an indistinguishable growth rate from control vehicle, as evident from the ratio of the **2**-treated worms reached to adulthood L4 and L3 stages, that were similar to the control group (Figure 6A). Even when the dose of compound **2** was brought up to 4.0 mM, worms still developed normally (data not shown). Insufficient food intake could lead to decrease fat storage, for instance, *eat-2* mutants have a malfunctioning feeding organ, and one of their phenotypes is a pale appearance with less fat store in the intestine [21]. To rule out the possibility that compound **2** caused caloric restriction on worms, food consumption rate was measured. It turned out that the **2**-treated worms ingested 1.5 fold more OP50 bacteria over the control in a given time period (Figure 6B). Of all checked phenotypes, only fecundity was influenced by the treatment of compound **2**. It decreased the brood size of N2 worms by 24% (Figure 6C). The reason for this particular outcome might lie in the fact that the germ-line demands a significant amount of lipid during peak reproduction in *C. elegans* [22], while lipid was in short supply in **2**-treated worms. Therefore, the lipid flux destined for reproductive system shrank. Hence, the gametogenesis will not develop full potential and larvae which normally harbor larger amounts of lipids and yolk inclusions fail in development. These data indicated that compound **2** was with low toxicity toward *C. elegans*.

### 2.4. Compound ***2*** Augmented Energy Catabolic Rate

Unbalanced lipid metabolism tends to give rise to abnormal fat storage in both mammal and *C. elegans* [21]. Nonetheless, they were leaner to accumulate less fat in their adipose tissue [23,24]. In order to verify whether compound **2** impacts on the energy flux processes in vivo, the ^13^C isotope labeling approach was utilized to detect lipid flux [25]. Under regular culture conditions, 7% palmitic acid (C16:0) of worms is synthesized de novo from an acetyl-CoA moiety, and fatty acids with a longer chain like stearic acid, oleic acid, linoleic acid and C20 polyunsaturated fatty acids are produced by elongation and desaturation from palmitic acid (C16:0). Thus, checking the rate of palmitic acid synthesis would provide a viewing window to the overall fat synthesis. The molecular weight of C16:0 isotopomers ranged from *m/z* 270 to *m/z* 286 on the mass spectrum. Upon the treatment of *C. elegans* by compound **2**, the quantities of ^13^C labeled C16:0 isotopomers within this molecular weight range were similar to that of the control group (Figure 7A). These results indicated that compound **2** did not intervene in the process of lipogenesis in worms. The conversion rate of palmitic acid to acid-soluble metabolites represents the level of β-oxidation of fat in N2 worms. To analyze energy efflux, the method described in the literature [26] was utilized to monitor the conversion rate of ^14^C labeled palmitic acid to acid-soluble metabolites. Compared to control worms, compound **2** enhanced the fat turnover rate in worms by twofold over the control (Figure 7B). This data suggested that compound **2** interfered the fat regulatory machinery in vivo via tipping the balance in favor of speedier fat mobilization.

For further investigation of the molecular mechanism by which compound **2** enhanced the rate of lipid mobilization in vivo, real-time PCR analysis was performed to measure the response of a panel of 96 metabolism-related genes after the worms were exposed to compound **2** (Table 2). The expression of three fat β-oxidation-related genes was altered by compound **2**, of which carnitine palmitoyl transferase gene (*cpt-3*) (located in the mitochondria) was upregulated by 5.6-fold and *acdh-1* and *acdh-2* which encode short chain acyl-CoA dehydrogenase decreased by 4.5-fold and 4.7-fold respectively. Gene *cpt-3* helped to mobilize fat stores for energy production by enhancing the flow of fatty acids into mitochondria for β–oxidation [27]. It was noted that human and mice with a deficiency of *acdh-1* and *acdh-2* resulted in the enhancement of intracellular C2-C6 fatty acids, while short chain fatty acids stimulate lipid β-oxidation and inhibit de novo lipid synthesis [28]. The experimental data suggested that compound **2** stimulated lipid β-oxidation and inhibited de novo lipid synthesis via the inhibition of *acdh-1* and *acdh-2.* In addition, compound **2** caused a significant down-regulation of SCD (*fat-7*) with 5.8-fold and up-regulation of the fatty acid binding proteins *lbp-8* (6.8-fold up) and *lbp-7* (3.3-fold up). However, no appreciable alternation was observed to the expression of NHR genes (*nhr-23, nhr-49, nhr-66*, *nhr-80, mdt-15*), which are important for energy homeostasis in *C. elegans* [29,30]. In addition, the expression levels of the genes for lipolysis, TCA cycle, de novo fatty acid synthesis and glucose metabolism were weakly affected by compound **2**. It is noted that *cpt-3, achd-1, acdh-2, fat-7* and *lbp-8* are fasting response genes, but here in **2**-treated worms, *lbp-8* changed in an opposite way to that in fasting worms, *lbp-7* was dramatically up-regulated instead of unchanged upon fasting. LBP-7, LBP-8 and other members of lipid binding proteins are predicted to function as intracellular free fatty acid transport, although their precise role in *C. elegans* is largely unknown. In sum, compound **2** performed a pronounced role in the promotion of fatty acid β-oxidation without inducing an overall fluctuation in fat metabolic gene workflow, but selectively enhanced and inhibited distinctive category genes.

### 2.5. Compound ***2*** Elicited the Fat-Reducing Effects Partially via Fat-5

Since compound **2** could decrease the lipid mass and accelerate the β-oxidation rate of energy metabolism in *C. elegans*, it is worthwhile to further investigate the underlying molecular mechanism by harnessing the power of *C. elegans* genetics. Basically, the core enzymes, proteins or pathways that regulate energy metabolism are well conserved across phyla. To verify whether compound **2** could retain its fat-reducing effect in worms with deficient genes that are responsible for modulating energy metabolism, the mutants in *fat-7, fat-6, daf-2, daf-4, daf-3, tph-1*, and *sbp-1*, which were susceptible to fat reduction, were detected. The experimental results showed that **2**-treated mutants had a similar decreasing degree of fluorescence levels as that of **2**-treated N_2_ (with 32.2% less Nile red staining) (Figure 8A). *aak-2* mutant, an energy sensor which encoded a catalytic subunit of AMP-activated kinase (AMPK), showed marginal resistance to compound **2**. Among all the tested mutants, *fat-5* mutants kept 66.6% Nile red fluorescence. The fat-reducing effect induced by compound **2** was partially abrogated in *fat-5* mutants, suggesting that FAT-5 is a potential target of compound **2** for fat-reducing effects. The association between AMPK and FAT-5 for the action of compound **2** required to be confirmed by further experiments. 

*C. elegans* possessed three SCDs (FAT-5, FAT-6, FAT-7) for the synthesis of monounsaturated fatty acids. FAT-5 is responsible for the conversion of palmic acid (C16:0) to hypogeic acid (C16:1 n7), while ELO-2 helped to convert C16:1 n7 to C18:1 n9 for the chain elongation. In contrast, FAT-6 and FAT-7 preferred to use stearic acid (18:0) as substrate [31]. Based on the feature of enzyme substrate specificity, it was possible to discern on what the particular SCD that compound **2** targeted. After the treatment by compound **2**, worm’s lipids were extracted and relative FFA composition was analyzed by gas chromatography/mass spectrometry (GC/MS). As shown in Figure 8B, the relative level of C16:1 n7 and C18:1 n9 in **2**-treated worms were significantly cut down by 39.0% and 16.6% respectively. This data echoed the results in the literature [32] that performed in fat-5 (tm420) mutant, and the decreased desaturation index (C16:1 n7/C16:0) indicated an attenuated FAT-5 enzyme activity [33]. Fatty acid (C18:2 n6) is the production of DAF-6 and DAF-7, but its relative level was marginally increased in **2**-treated worms. Although we did not have positive statistics values regarding the level of C16:0, a slight build-up of this substrate in **2**-treated samples in each of our experimental trials was observed. The GC/MS data supported the speculation that compound **2** may impair FAT-5 enzyme so that it is less potent in catalyzing the conversion of C16:0 to C16:1 n7. Our results are also consistent with previous studies that deficient SCDs caused altered FFAs profiles [34,35].

## 3. Discussion

Since lipid content modulation is a physiologically relevant phenotype that is the outcome of an array of integrated pathways, it is likely that compound **2** also works on an additional set of targets or effector proteins; impaired FAT-5 probably leads to a knock-on effect in an intricate feedback and compensation energy system. In this work, the doses of compound **2** used for the fat phenotypes in worms were relatively high. However, the ultra performance liquid chromatography (UPLC) peak of compound **2** in the worm body was undetectable when the worms were soaked to 1 mM of compound **2** (Appendix A), the maximum dose for fat-reduction. Since the limit of detection of compound **2** in the UPLC chromatography was 2 μM per 10 μL of injection, it was supposed that the intake of compound **2** by worms was less than 2 μM.

## 4. Materials and Methods

### 4.1. General Information

Optical rotations were measured with Autopol IV Rudolph Automatic polarimeter (Rudolph Research Analytical, Hackettstown, NJ, USA). IR spectra were recorded on a Nicolet Nexus 470 FT-IR spectrometer (Thermo, Boston, MA, USA). NMR spectra were measured on Bruker Avance-500 spectrometer (Bruker Cryomagnet, Fällanden, Switzerland) using tetramethylsilane (TMS) as an internal standard. HRESIMS spectra recorded on Bruker P-SIMS-Gly spectrometer (Bruker, Karlsruhe, Germany). ECD spectra were measured on a JASCO J-810 spectropolarimeter (Jasco, San Diego, CA, USA). Column chromatography was performed on silica gel (Qingdao gel C-200 and C-300, Qingdao Marine Chemical Co., Qingdao, China), HPLC was performed on an Alltech 426 apparatus with a UV detector. YMC-Pack C_8_ column (ODS, 10 mm × 250 mm, for reversed-phase) was purchased from YMC Europe GMBH (Schoettmannshof, Germany). Nile red and other reagents were purchased from Sigma-Aldrich (Burlington, CO, USA).

### 4.2. Mutation and Cultivation of Streptomyces Nitrosporeus

The information for sample collection and the species identification of *S. nitrosporeus* (T14-24) was referenced from the literature. For the chemical-induced mutation, spores of the stain (T14-24) were collected and diluted with Tris–HCl buffer (50 mM, pH 6.0) to distribute the spores in 10^7^ spores/mL. The spores suspension (2 mL) was treated with NaNO_2_ (1M, 1 mL) and acetic acid buffer (0.1 M, 7 mL, pH 4.6) for 15 min. Na_2_HPO_4_ buffer (0.07 M, 9 mL, pH 8.6) was added to terminate the reaction. Spore suspension (100 μL) was transferred to an agar plate and cultured at 30 °C for 7 days. The selected mutated strain encoded YBH10-5 was inoculated in 500 mL Erlenmeyer flasks containing liquid medium (150 mL, soluble starch 2 g, KNO_3_ 0.1 g MgSO_4_ 0.05 g, K_2_HPO_4_ 0.05 g, FeSO_4_ 0.001 g, urea 0.03 g, extracting solution of soil 20 mL, distilled water 80 mL, pH 6.8), followed by shaking incubation at 100 rpm for 14 days at 28 °C.

### 4.3. Extraction and Isolation

The fermented broth (40 L) of YBH10-5 strain including supernatant and mycelium was extracted with hexane, EtOAc, while the water-soluble fraction was concentrated in vacuum. The hexane and EtOAc extracts were also concentrated in vacuum to obtain fractions FA and FB, while the concentrated water-soluble fraction was extracted by MeOH to yield fraction FC. The three fractions (FA, FB and FC) were assayed for lowering lipid effects in HepG2 cells, while only FB (EtOAc) fraction showed a positive effect. The EtOAc fraction (17 g) was subjected to VLC silica gel column using a stepwise gradient elution of petroleum ether/acetone/MeOH, to yield six fractions (FB1-FB6). Following the same assay protocol as that for FA-FC, FB3 showed stronger lowering lipid effect than other fractions. FB3 (1.5 g) was subjected to extensive column chromatography including the semi-preparative HPLC separation as described in the literature [10] resulted in the isolation of **1** (3.2 mg), **2** (2.5 mg), **3** (20 mg), **4** (3.0 mg), **5** (5.0 mg), **6** (6.0 mg), **7** (4.0 mg), **8** (4.0 mg), **9** (3.5 mg), **10** (5.3 mg) and **11** (6.2 mg).

### 4.4. Spectroscopic Data of Nitrosporeunol H (***1***)

Purplish red amorphous, ${\left[\alpha \right]}_{D}^{20}$ −28 (*c* 0.3, MeOH), UV (MeOH) λ_max_ (log ε) 201 (2.59), 262 (0.72), 338 (0.10) nm, IR (KBr) ν_max_ 3446, 2963, 2921, 1653, 1629, 1453, 1234, 1023 cm^-1^, ^1^H and ^13^C NMR data, see Table 1, HRESIMS *m/z* 647.4106 [M − H]^−^ (calc. 647.4100).

### 4.5. Worm Strains

Wild type *C. elegans* worms (N2 Bristol strain) and the worm mutants (aak-2(ok524)X, daf-2(e1370)III, daf-3(e1376)X, daf-16(m26) I, daf-4(e1364)III, fat-5(tm420), fat-6(tm331), fat-7(wa36),tph-1(mg280)II, tub-1(nr2044)II, sbp-1(ep79) III, nhr-49(nr2041)I) were provided by Celegans Genetic Center (the University of Minnesota, Minneapolis, MN, USA). Worms were cultured and scored at 20 °C.

### 4.6. Phenotype Assay

Experiments were performed on 96-well plates, each well contained 100 μl NGM agar. The compound solution was added droplet on the surface of OP50 seeded well to make a final concentration of 500 μM (or otherwise stated) with a 200 nM fluorescent dye Nile red. Sit at room temperature for 2–4 h for chemical distributing well before using. All experiments were duplicated three times. For Nile red phenotype scoring, L1 larvae were grown until most worms reached to L4 or young adult stage, while worms were picked and transferred to a new seeded plate for photographing. Immediately prior to image acquiring, anaesthetized worms with NaN_3_, All images were shot on Olympus fluorescent microscopy with same settings. For the post-embryonic developmental assay, synchronized L1 stage larvae were transferred to plate. After 2–3 days, the number of worms in each developmental stage (namely, young adult, L4, L3 and larva) was counted by visual examination using a dissection microscope. For brood size counting, larvae were placed singly onto new wells containing either testing compound or DMSO to grow until they matured and laid the first few eggs, the hermaphrodites then were transferred daily onto fresh wells to prevent overcrowding until egg laying ceased. The progeny on each well was counted. The brood size of the average of 10–15 worms per condition was reported. For food consumption assay, synchronized L1 larvae were grown in wells with compound or control (DMSO) till most worms reached to L4 or young adult stage. Young adults (10) or L4 worms (rinse with M9 buffer three times to remove stained bacteria before transferring) were transferred to fresh wells which were seeded with 1 × 10^9^ cfu OP50. After 12 h, left bacteria were collected with 100 μL M9 buffer to rinse three times, then to make 1/100, 1/1000,1/10,000 serial dilution on an agar plate. In next day, the number of individual colonies on the plate was counted with multiple degrees of dilution to get the CFU number.

### 4.7. Oil Red O Staining of Lipid

HepG2 cells were maintained in DMEM medium supplemented with 10% fetal bovine serum and penicillin/streptomycin (100 μg/mL). The cells with 70%–80% confluence were incubated in DMEM plus oleic acid (100 μM) for 12 h, then were treated with compound 2 (10 μM) with DMEM/100 μM oleic acid as a blank for additional 6 h. Subsequently, the cells were subjected to oil-red O staining as described previously [16]. Each experiment (*n* = 8 for oil-red O staining) was repeated in triplication. Steatosis modeling in HepG2 cells and lipid staining with Oil Red O were performed as described in the literature [16].

### 4.8. Triglyceride Colorimetric Assay

Synchronized L4 and young adult worms are collected and rinsed three times with M9 buffer, an aliquot was kept for BCA protein measurement (Piece), the rest of the worms were processed for triglyceride quantification using a commercial kit (Mountain View, CA, USA) by following kit’s instruction.

### 4.9. Fluorescence Quantification

Image J was used for quantitation of fluorescent density, Nile red stained lipid mostly located at intestine and the anterior part usually was brighter than posterior. The area for integrating fluorescent density measurement was selected from most anterior front to vulva of the intestine. Six L4 worms of each well were randomly picked for measurement. Images for Image J analysis were shot at 20× magnification.

### 4.10. QPCR Analysis

Synchronized L4/young adult population of worms were collected by washing off plates with M9 buffer (three times) at room temperature. Total RNA was prepared from those worms using Trizol™ reagent, and then the cDNA was produced from 4 μg of total RNA in a 100 μL volume reaction using cDNA preparation kit (Promega). PCR was carried out with SYBR-Green PCR kit (A6001, Promega) (Qiagen, Duesseldorf, Germany) in a 25 μL reaction using Biorad iCycler (Bio-Rad, Boston, MA, USA). The relative level of each transcript was calculated by the comparative Ct method. tbg-1 (gamma-tubulin) served as a reference gene. Gene list for analysis was those described by Gilst [36], plus 6 nuclear hormone receptor (nhr-49) or co-factor genes and genes of nine fatty acids transporting proteins. qRT-PCR primers were designed using NCBI Primer-Blast (http://www.ncbi.nlm.nih.gov/tools/primer-blast), primer sequences were available upon request.

### 4.11. GC/MS Spectrometry for Fatty Acid Profile Analysis

Synchronized L4/young adult population of worms were collected and rinsed three times with cold M9 buffer in 15 mL conical tubes, letting worms settle to form sediment without centrifugation in order to clear off as much bacteria as possible. Packed worms were stored at −80 °C for later lipid analysis. After removing as much water as possible, lipids were extracted and methyl esterification was performed simultaneously by sonicating the worm samples in 2.5% H_2_SO_4_ in MeOH for 1 h and followed by incubating at 80 °C for the other 1 h. GC/MS spectra were collected on an HP 6890 gas chromatograph outfitted with a J and W DB-XLB (Agilent Technologies, Palo Alto, CA, USA) column. The mass spectrometer was an HP MSD 5973 (Agilent, Palo Alto, CA, USA) and data were analyzed using Chemstation version A.03.00 software (Agilent Technologies, Palo Alto, CA, USA). Peaks were assigned using fatty acid standards.

### 4.12. Determination of β-oxidation

The protocol was adapted from the literature [26] with minor modification. Briefly, 20,000–30,000 synchronized L4/young adult worms were collected and rinsed three times with M9 buffer to remove carried bacteria OP50, an aliquot was stored at −80 °C for subsequent protein determination (Pierce), the rest worms were incubated with ^14^C-palmitic acid (6 μCi) in BSA for 2 h with agitation. Radioactivity of the supernatant was determined by liquid scintillation (PerkinElmer) after precipitating with perchloric acid. The experiments were performed in triplicates two independent times.

### 4.13. Internal Compound Concentration

Synchronized L4 N2 worms were soaked in a solution of 500 μM compound for 4 h with gentle agitation, while worms were pelleted and washed for three times with cold M9 buffer. The supernatant was removed carefully such that as little liquid left as possible, one-third of the packed worms was transferred to another tube and two volumes of RAMP lysis buffer were added, vortex wells were stored at −80 °C for late protein measurement. 2 volumes of lysis buffer (NaCl, Tris, protease K SDS%) were added into the rest worms, 2 volumes of lysis buffer (NaCl, Tris, protease K SDS%) were added in vortex well at 55 °C for 60 min.

### 4.14. Statistical Analyses

All results were statistically analyzed using GraphPad Prism 5 (version 5, GraphPad Software, San Diego, CA, USA) and reported as mean ± SD, Significant differences were considered when *p* < 0.05.

## 5. Conclusions

In conclusion, the present work uncovered a natural farnesylquinone (**2**) to reduce fat mass in *C. elegans* without obvious deleterious effects on the general health of worms and mammal cell line HepG2. Compound **2** was supposed to interfere the energy homeostasis in *C. elegans* by accelerating the β-oxidation rate and to modulate the expression of six lipid metabolism genes, restrains the activity of one subtype Δ9 desaturase FAT-5 to change the FFA profile in live worms. These data suggested that compound **2** was a promising natural scaffold to be developed as a novel SCDs isoform-specific inhibitor with higher potency and low toxicity after chemical modification and optimization.

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
