# Peer review of "Reducing Effect of Farnesylquinone on Lipid Mass in C. elegans by Modulating Lipid Metabolism"

_marinedrugs, 2019, doi:10.3390/md17060336_

Round 1
Reviewer 1 Report
Jia et al identified 11 compounds from ethanolic extract of Streptomices nitrosporeus culture broth and tested their ability at modulating lipid metabolism in vivo, also providing a possible mechanism.
I found the manuscript very interesting, with a clear experimental plan. However, I think that the manuscript needs some modifications before I would recommend its acceptance.
My concerns and comments are listed below:
1. Authors name the compounds with numbers through all the manuscript. I understand this is the easier way to write but I would always refer to the compound as “compound 2” and not only as “2”.
2. Authors discuss most of the results already in the Results section thus I think they should merge Results with the Discussion section.
3. The manuscript should be checked carefully for language and style.
Moreover, some sentences should be rephrased to make them understandable.
For example, in lines:
· 52-55
· 126-127
· 130-132
· 162-163
· 207-209
· 252-253 (verb missing)
· 275-278
4. Results section, figure 4, lines 145-146. Authors write that compound 6 is less efficient at decreasing the fat mass in worms, as visible also in the figure showing no significant decrease of Nile red fluorescence. But then they write “decrease of efficacy with the disappearance of the Nile red phenotype”. Probably this sentence is only poorly written and needs to be rephrased but it seems that they are stating two different conclusions.
5. Results section, figure 5.
· Panel B, add significance
· Panel C, which concentration of compound 2 has been used?
· Panel D, which is the rationale for using 10 mM of compound 2?
6. Results section, figure 8. It would be good to add in the bar graph in A panel also the fluorescence obtained for the untreated worms.
7. Materials and Methods section, paragraph 4.7. Describe briefly the experimental procedure performed.
8. It would be good to have an Abbreviations paragraph.
Author Response
Answer to reviewer-1
(1) Authors name the compounds with numbers through all the manuscript. I understand this is the easier way to write but I would always refer to the compound as “compound 2” and not only as “2”.
Response: I agree the comment to put the term “compound” before the number which represents the compound. I added “compound” before the numbers for the relevant compounds through full text and also experimental section.
(2) Authors discuss most of the results already in the Results section thus I think they should merge Results with the Discussion section.
Response: Thanks for the comment. In the “Discussion” section, I deleted the sentences which have been mentioned in the “Result” section, but maintain the sentences which described that the dose that worm intake differed from the doses what we used , indicating more less of the dose that worms intake. In the “Conclusion” section, we summarized the achievement in the full text.
(3) The manuscript should be checked carefully for language and style. Moreover, some sentences should be rephrased to make them understandable.
For example, in lines: 52-55, 126-127, 130-132, 162-163, 207-209, 252-253 (verb missing), 275-278.
Response: I checked full text carefully and revised the sentences based on the referee’s comment and the errors or typos what we found.
The sentences in lines 53-55 are revised to “More than 400 genes in C. elegans directly attend lipid synthesis and catabolism, and play the roles as pivotal energy sensors or foraging regulators, while energy mobilization and fat storage have counterparts in mammals [13].”
The sentences in lines 126-127 are revised to “C. elegans (worm) deposits its fat mainly in intestine, but it lacks adipocytes that are found in all mammals.”
The sentences in lines 130-132 are revised to “In preliminary experiments, all compounds were tested for the effects to regulate fat in C. elegans, based on the Nile red fluorescence intensity that represents lipid droplets in N2 worms.”
The sentences in lines 162-163 are revised to “To further ascertain the fat phenotype as induced by compound 2 to be related to its fat-lowering effects, the triglyceride level of 2-treated worms was measured.”
I deleted the sentences in lines 207-209 since we did not do the relevant experiment in mice.
The sentences in lines 252-253 are revised to “However, there is no appreciable alternation to the expression of NHR genes (nhr-23, nhr-49, nhr-66, nhr-80, mdt-15), which are important for energy homeostasis in C. elegans [29,30].”
The sentences in lines 275-278 are revised to “Since compound 2 could decrease the lipid mass and accelerate the β-oxidation rate of energy metabolism in C. elegans, it is worthwhile to further investigate the underlying molecular mechanism by harnessing the power of C. elegans genetics.”
(4) Results section, figure 4, lines 145-146. Authors write that compound 6 is less efficient at decreasing the fat mass in worms, as visible also in the figure showing no significant decrease of Nile red fluorescence. But then they write “decrease of efficacy with the disappearance of the Nile red phenotype”. Probably this sentence is only poorly written and needs to be rephrased but it seems that they are stating two different conclusions.
Response: the sentence is revised to “Oxidation of the quinone core such as compound 6 led to the disappearance of the Nile red phenotype, implying that quinone or its phenol ring is the essential core to play the fatty-lowering role.”
(5) Results section, figure 5.
Panel B, add significance
Panel C, which concentration of compound 2 has been used?
Panel D, which is the rationale for using 10 mM of compound 2?
Response: In Figure 5, the significance in Panel B is added; (2) The concentration of 2 used for the experimental is 1 mM, that is added under the footnote; (3) In Panel D, the dose of 2 is 10 μM that is the maximum dose used to test the lipid-lowering effect of compound in HepG2 cells based on the bioassay regulation in our laboratory.
(6) Results section, figure 8. It would be good to add in the bar graph in A panel also the fluorescence obtained for the untreated worms.
Response: The bar graph in Panel A is added in the revised version. In Panel A, the fluorescence of control group (DMSO) of worms is presented for the untreated worms.
(7) Materials and Methods section, paragraph 4.7. Describe briefly the experimental procedure performed.
Response: The experimental procedure performed for O red staining HepG2 cells is provided in the experimental section.
(8) It would be good to have an Abbreviations paragraph.
Response: Based on the editing style of Marine Drugs, no section is afforded to locate Abbreviations paragraph. I suggest editor to confirm whether we need to provide this paragraph.
Reviewer 2 Report
Xihua Jia et al., shown the reducing effect of Franesylquinone on lipid mass in C.elegans – by modulating lipid metabolism. This article is somewhat well written, and it fits into the scope of this journal. Please find comments below,
1. In C. elegans both lipid droplets and yolk particles contain mainly phospholipid species and contain TAGs, but very little cholesterol or cholesterol esters. Does the author claim that Farnesylquinone reduces the TAGs content.
2. According to Lipidomics study - The fatty acid composition of C. elegans TAG revealed a high content of dietary fatty acids in lipid droplets. TAG fractions contained relatively smaller proportions of de novo synthesized fatty acids compared to phospholipid fractions. Does the Franesylquinone will have a similar effect in other animal models?
3. Numerous studies have shown that cholesterol levels are very-low in C. elegans membranes and the primary function of cholesterol is for the production of steroid hormones regulating growth and reproduction, rather than as structural components of cellular membranes. This is entirely different in humans and other animal models – How does the author interpret this with the current study
4. Many of the adverse health effects of excess fat accumulation in humans are unlikely to occur in C. elegans. Does the Franesylquinone will have a similar effect?
Author Response
Answer to reviewer-2
(1) In C. elegans both lipid droplets and yolk particles contain mainly phospholipid species and contain TAGs, but very little cholesterol or cholesterol esters. Does the author claim that Farnesylquinone reduces the TAGs content.
Response: In our experiments, we measured the TAGs content of compound 2 treated N2 worms, and found out that treated worms had 24% less TAGs compared to that of DMSO control group. This result indicated that farnesylquinone could reduce the TAGs in C.elegans animal model.
(2) According to Lipidomics study - The fatty acid composition of C. elegans TAG revealed a high content of dietary fatty acids in lipid droplets. TAG fractions contained relatively smaller proportions of de novo synthesized fatty acids compared to phospholipid fractions. Does the Franesylquinone will have a similar effect in other animal models?
Response: Although C. elegans obtain most of the fatty acids from bacterial diets, ~7% C16:0 comes from de novo synthesis from acetyl CoA , deficiency in genes responsible for de novo synthesis of fatty acids in vivo, such as pod-2 (acetyl-CoA carboxylase) and fasn-1 (Fatty Acid SyNthase) leads to fat reducing phenotype in worms. Stearate (C18:0), oleate (C18:1n9), linoleate (C18:2n6), and several 20 carbon polyunsaturated fats are not available in the common laboratory diet (E. coli [OP50]) and must be obtained by elongation and desaturation of C16:0. The production of longer monounsaturated fatty acids is controlled by desaturases (FAT-1 through FAT-7) and elongase (ELO-1 and ELO-2), The fat-6;fat-7 double mutants have 20% less TAG stores compared to wild type .in our study, compound 2 displayed its activity partially via the inhibition of FAT-5, which is a counterpart of mammal Δ9-desaturase(SCD), The fat-storage phenotypes observed in C. elegans parallel those in Δ9-desaturase deficient mice, which have low fat stores, increased metabolic rate and increased expression of fatty acid oxidation genes. We tested compound 2 (10 μM) in HepG2 steatosis model and observed similar fat-reducing phenotype.
We are sorry that we did not test other animal models such as mice due to the limited amount of samples.
(3) Numerous studies have shown that cholesterol levels are very-low in C. elegans membranes and the primary function of cholesterol is for the production of steroid hormones regulating growth and reproduction, rather than as structural components of cellular membranes. This is entirely different in humans and other animal models – How does the author interpret this with the current study.
Response: C. elegans can not produce cholesterol by themselves, and the content level of this particular lipid type is very low. As reviewer said, its function is pretty much different compared to that in mammals. It is well documented that fat storage is closely related with fatty acid synthesis, elongation and desaturation. In our study, we focused on the fatty acids profiles changes upon compound treatment. Altered monounsaturated FFAs composition is one of the mode of action of compound 2. This mechanism of action is similar to that of mouse harbors SCD mutation.
In addition, the core molecular mechanisms to regulate lipid metabolism are well conserved across phyla. In present work, we mainly focus on the mitochondrial β-oxidation rate and the transcriptional level of energy metabolism genes in worms that were treated by compound 2. Additional data to compare the effects of compound 2 between C. elegans and other animals are required after we accumulated sufficient amount of samples by re-isolation of large scale fermentation.
(4) Many of the adverse health effects of excess fat accumulation in humans are unlikely to occur in C. elegans. Does the Franesylquinone will have a similar effect?
Response: Mounting evidence indicated that the pathways as well as genes that regulate the energy homeosis is well conserved in C .elegans. In this study, we take advantage of the power of C .elegans genetic tools to dissect the mode of action of compound 2, its in vivo target is FAT-5 which is an ortholog of mammal SCD, mice with SCD deficiency is slimmer. Of all checked phenotypes, only fecundity was influenced by the treatment of compound 2. It decreased the brood size of N2 worms by 24%. The reason for this particular outcome might lie in the fact that the germ-line demands a significant amount of lipid during peak reproduction in C. elegans, while lipid was in short supply in 2-treated worms. Therefore, the lipid flux destined for reproductive system shrank. These data indicated that compound 2 was with low toxicity toward C. elegans. The manuscript provided the primary data to understand the mode of action of franesylquinone to induce the fat reduction. More detail experiments should be designed to test the toxicity and side effects in mammal models in future after we accumulate sufficient amount of compound. This will be reported in additional work.